

# Foraging efficiency, social status and body condition in group-living horses and ponies

Sarah L. Giles[1], Pat Harris[2], Sean A. Rands[3] and Christine J. Nicol[4]

[1] School of Veterinary Sciences, University of Bristol, Langford, North Somerset, UK
[2] Equine Studies Group, WALTHAM Petcare Science Institute, Melton Mowbray, Leicestershire, UK
[3] School of Biological Sciences, University of Bristol, Bristol, UK
[4] Royal Veterinary College, Hatfield, UK

## ABSTRACT

Individual animals experience different costs and benefits associated with group living, which may impact on their foraging efficiency in ways not yet well specified. This study investigated associations between social dominance, body condition and interruptions to foraging behaviour in a cross-sectional study of 116 domestic horses and ponies, kept in 20 discrete herds. Social dominance was measured for each individual alongside observations of winter foraging behaviour. During bouts of foraging, the duration, frequency and category (vigilance, movement, social displacements given and received, scratching and startle responses) of interruptions were recorded, with total interruption time taken as a proxy measure of foraging efficiency. Total foraging time was not influenced by body condition or social dominance. Body condition was associated with social dominance, but more strongly associated with foraging efficiency. Specifically, lower body condition was associated with greater vigilance. This demonstrates that factors other than social dominance can result in stable differences in winter body condition.

## INTRODUCTION

Social behaviour can influence energetic reserves and subsequent body condition. Previous modelling studies have outlined the potential importance of social effects on foraging behaviour (bouts of biting, chewing and swallowing interrupted by relocation movements) in determining body condition in group living animals (*Houston & McNamara, 1999*; *Rands et al., 2003*, *2004*, *2006*, *2008*) and also the role of dominance behaviours in determining resource access and subsequent body condition (*Clark & Ekman, 1995*; *Stillman, Goss-Custard & Caldow, 1997*; *Rands et al., 2006*). Thus, the foraging success of individual animals in social groups may be partly influenced by their social status. However, few of these predictions have been investigated empirically in socially-foraging herbivores and the relationship between herd behaviours, dominance and body condition is not fully understood.

Corresponding author
Christine J. Nicol, cnicol@rvc.ac.uk

In a socially foraging herbivore the benefits of group living outweigh the costs (*Krause & Ruxton, 2002*). Individual animals living within groups follow behavioural rules which allow them to function as a social unit (*Hemelrijk, 2002*; *Rands, 2011a*, *2011b*). These rules are likely to depend upon both aspects of their own body condition (such as energetic reserves) and also the actions of other individuals within the group (*Houston & McNamara, 1999*; *Rands et al., 2003*, *2008*). Rules governing social interaction (e.g. dominance) may be important for a well-functioning group in terms of minimising costly conflict over resources (*Krause & Ruxton, 2002*).

*Rands (2011b)* considered a game theoretical framework to explore how the rules used by individuals with different dominance ranks could evolve, assuming these individuals paid attention to the ranks and energetic state of both themselves and the individual that they were interacting with. This model, and a companion simulation exploring the rules of thumb generated (*Rands, 2011a*) demonstrated that both energetic state and social status are important for determining the behaviour of co-foraging individuals. Furthermore, individual-based simulations (*Rands et al., 2004*, *2006*) demonstrated that including an additional effect of dominance that led to subordinates having reduced access to food could lead not only to dominant individuals having higher energetic reserves than subordinates, but also subordinate individuals increasing their activity.

We aimed to assess whether this framework was useful in understanding the foraging behaviour of the horse. We were particularly interested to determine whether dominant animals had higher body condition and whether subordinate individuals showed increased activity in line with model predictions. Horses are generalist herbivores with sophisticated social capacities. Free-ranging feral and primitive Przewalksi's horses spend a high proportion of each day foraging (52%, *Berger et al., 1999*; 68%, *Lamoot & Hoffmann, 2004*; up to 75% daylight and 53% nocturnal, *Mayes & Duncan, 1986*) maintaining a high daily intake of plant material by grazing (or browsing) interrupted by frequent walking (*Houpt, 2005*). Accelerometry studies find similar proportions of time spent foraging by domestic horses kept on pasture (61% daylight, 47% nocturnal, *Maisonpierre et al., 2019*). Horses form strong affiliative bonds with familiar companions, but aggressive encounters and subtle threats, are also a common feature of equine social structure, particularly when resources are limited (*Mills & Redgate, 2017*). The current study was conducted under winter conditions where pasture availability was limited and a degree of competition for supplementary forage was evident. The situation applies commonly for domestic horses (kept for a variety of reasons including as companion animals or as conservation grazers (*Gilhaus & Hoelzel, 2016*) during winter periods within temperate zones). Understanding the factors that drive large inter-individual differences in body condition when group-living horses are kept during winter (*Ingólfsdóttir & Sigurjónsdóttir, 2008*; *Giles et al., 2015*; *Yngvesson et al., 2019*) is an important goal. It has been estimated that around a third of outdoor living horses and ponies within the UK are obese (*Giles et al., 2014*; *Robin et al., 2015*) but rates of obesity can reach 70% in some populations (*Menzies-Gow, Harris & Elliott, 2017*). It is timely to study the social factors influencing body condition in horses to reduce obesity prevalence and associated metabolic disease.

Previous empirical studies in horses have demonstrated that higher ranking individuals spend more time eating hay and have a higher body condition during the winter (*Ingólfsdóttir & Sigurjónsdóttir, 2008*; *Giles et al., 2015*) but have not examined the mechanisms behind this association.

This study advanced our previous work by examining situations where bouts of foraging on supplementary forage were *interrupted* for reasons including anti-predator vigilance and startle responses (*Goodwin, 1999*), displacement interruptions directed towards or received from other group members (*Appleby, 1980*; *Rands et al., 2006*) or short movements between foraging locations (*Duncan, 1980*). We examined the duration, frequency and type of interruption to the foraging behaviour of individual horses and ponies (hereafter termed 'horses') living in social herds. The total time attributed to interrupted foraging was considered as a proxy measure of foraging efficiency (the ratio of energy gained over energy expended during foraging).

An important precursor to analysing foraging efficiency was understanding any differences in overall time spent foraging. We measured overall time spent foraging to check that individuals with a lower foraging efficiency didn't simply compensate by spending more time foraging. A unique feature of the study was the inclusion of measures of social status and body condition, enabling the assessment of associations not previously examined in foraging herbivores. Predictions suggest that subordinate individuals may suffer more displacement than dominant conspecifics (*Goss-Custard et al., 1995*; *Stillman, Goss-Custard & Caldow, 1997*; *Stillman et al., 2000*; *Rands et al., 2006*), reflected in increased displacement interactions and subsequent movement within foraging bouts. Dominant animals may also force subordinate conspecifics into more exposed foraging positions (*Ekman, 1987*; *Rands et al., 2004*) leading to a reduction in foraging efficiency due to a greater requirement for vigilance. In contrast, models predict that dominant individuals will be more efficient foragers, feeding in positions with lower interference, potentially leading to a greater energetic intake and overall body condition (*Ekman, 1987*; *Schneider, 1984*; *Rands et al., 2006*). A greater body condition may in turn allow a subsequent competitive advantage (*Rands, 2011b*; *Rands et al., 2006*).

Our aims were to:

i) Confirm an association between dominance rank (adjusted for herd size, see "Methods") and body condition.

ii) Assess whether adjusted dominance rank is associated with interruptions to foraging (as a proxy for foraging efficiency).

iii) Assess whether body condition is associated with interruptions to foraging (as a proxy for foraging efficiency).

iv) Use multivariate analysis to investigate the contextual factors (age, breed, sex, height, supplementary feeding) that might influence these associations.

v) Consider the applied implications of our findings for the management of domestic horses.

We predicted that foraging interruptions would be associated with both body condition and dominance status, and that subordinate individuals would, overall, have a reduced foraging efficiency compared with more dominant conspecifics and a lower body condition, as indicated in a previous study (*Giles et al., 2015*). This study goes beyond previous research to assess whether differences in foraging efficiency could plausibly be the mechanism linking dominance to body condition.

## MATERIALS AND METHODS

### Animals and ethical statement

The work was approved by the University of Bristol Animal Welfare and Ethical Review Board (University Investigation Number UB/10/049) and all methods were carried out in accordance with relevant guidelines.

The study sample was drawn from a population of outdoor, group-living horses based at Redwings Horse Sanctuary (UK), that had been living together for at least 3 months and had established social relationships. All of the individual animals were managed similarly, fed forage from identical sources, lived in outdoor environments and were not ridden, meaning that structured exercise could be removed as a potential confounding factor. Herds that included pregnant or lactating mares were not considered for the study. Twenty study herds were selected randomly from all remaining suitable herds within the sampling frame.

The policy of the sanctuary was to house horses in relatively compatible groups with shared characteristics. Thus, larger horses were housed in separate herds from smaller ponies, all stallions were housed in one 'bachelor' herd, while youngsters were also housed together, with the few horses under 1 year of age (three individuals) accompanied by older 'nanny' mares. Herd size was 2–10 (mean 6 ± 0.56 individuals). 116 individuals (84 ponies of height <148 cm, and 32 horses of height ≥148 cm) from within these herds were observed between 2 December, 2013 and 23 January, 2014. Ages ranged from 5 months to 32 years (11.83 ± 0.63 years). Breeds were native ponies (51.72%), native cobs (17.24%), lightweight horses (12.07%), heavy horses (5.17%), sports horse breeds (5.17%) and other (8.62%).

### Study period and horse management

The winter months were chosen for observation as natural food resources were at their minimum and therefore food based social interactions were likely at their highest due to the close proximity of individuals. All horses lived in an outdoor paddock environment for 24 h a day and were fed from circular hay feeders provided at a fixed ratio of feeder space (30 cm) per animal. Horses were fed twice daily with fresh hay replenished once at the start of morning observation (between 08:00 and 09:00) and once at the start of afternoon observation (between 11.30 and 13:00). Any uneaten hay remained in the hay feeder throughout the day. Twelve study horses received additional supplementary feed from a bucket once a day, and this was recorded as a potential confounder.

## Time spent foraging

Each study herd was observed for 6 h to assess overall time spent foraging, and interruptions occurring during foraging bouts, once during a 3 h morning session (08:00–09:00 until 11:00–12:00) and once during a 3 h afternoon session (11:30–13:00 until 14:30–16:00) on a different day within the same week, by a single trained observer. Due to the time of year, these times were chosen based on daylight hours.

Time spent foraging was recorded using scan sampling at 5 min intervals throughout each 3-h observation period. A random number generator was used to determine the order in which individuals were observed. Once this order was determined, all individuals were observed in sequence, in 5-min intervals. At each interval, it was recorded which individuals were foraging and which were not. Foraging was defined as the horse ingesting either hay or grass, with intermittent periods of the head down ingesting forage and the head up chewing this forage material. The horse could be foraging from either the hay feeder or eating grass (although the latter was rare as there was little grass available). The percentage of time spent foraging was then calculated based on the number of intervals that each individual was foraging within the full 6 h of observation per herd.

Alongside this, continuous 5 min focal animal observations were scheduled for each horse during each 3 h recording period. Each individual animal was independently observed for at least 20 min (4 × 5-min) in total. These observations were predominantly used to record foraging interruptions and social interactions (as detailed in "Foraging Efficiency—Duration and Frequency of Foraging Interruptions" and "Dominance Rank" below), however they were also used to more accurately estimate the total foraging time for each individual. If an individual was not foraging for more than 1 min during the 5-min observation period, it was considered to have stopped foraging. The number of minutes it had stopped foraging for were then subtracted from the total 5 min.

## Foraging efficiency—duration and frequency of foraging interruptions

During the continuous 5-min focal animal observations, described above, observations relating to foraging interruptions were also conducted. Interruption to foraging was defined as an activity that was short in duration (less than 1 min) and prevented the individual from selecting, biting or chewing hay or grass. Both the frequency and overall duration of any interruption was recorded and interruptions were categorised as one of the following:

*Vigilance:* Head raised from foraging and ears pricked in the direction of interest, the head is higher and the ears upright distinguishing vigilance from raising the head to chew.

*Movement whilst foraging:* a short movement resulting in a change in foraging location, either following a displacement by another individual or simply changing location at a walk.

*Displacements given*: interaction directed towards another individual, with the head outstretched and ears flat back against the head resulting in recipient raising head, or taking a step away in any direction.

**Table 1 Statistically significant univariable associations (*p* ≤ 0.05) using mixed effects linear regression, controlling for herd group and herd size as a random effects.**

| | Interruption behaviour variables | β | SE | 95% CI | Z | p |
|---|---|---|---|---|---|---|
| Adjusted dominance rank and Body Condition Score | | 0.66 | 0.29 | [0.09–1.24] | 2.27 | 0.023 |
| Body condition and foraging efficiency | | | | | | |
| Frequency | Total instances of interruptions | −0.77 | 0.29 | [−1.33 to −0.21] | −2.71 | 0.007 |
| | Instances of vigilance | −0.93 | 0.30 | [−1.52 to −0.34] | −3.09 | 0.002 |
| Duration | Total duration of interruptions | 0.08 | 0.04 | [−0.15 to −0.01] | 2.50 | 0.012 |
| Adjusted dominance rank and foraging efficiency | | | | | | |
| Frequency | Instances of moving whilst foraging | −0.85 | 0.30 | [−1.45 to −0.25] | −2.77 | 0.006 |
| | Instances of displacements received | −0.07 | 0.02 | [−0.11 to −0.03] | −3.62 | <0.001 |
| | Instances of displacements given | 1.36 | 0.33 | [0.71–2.01] | 4.12 | <0.001 |
| Duration | Total duration of interruptions | −0.02 | 0.01 | [−0.04 to −0.001] | −2.06 | 0.039 |
| Associations between interruption behaviour variables | | | | | | |
| Frequency of displacements received | | | | | | |
| | Instances of moving whilst foraging | 0.20 | 0.06 | [0.08–0.32] | 3.38 | 0.001 |
| | Instances of displacements given | −0.16 | 0.07 | [−0.29 to −0.02] | −2.30 | 0.021 |
| Frequency of displacements given | | | | | | |
| | Instances of moving whilst foraging | −0.16 | 0.08 | [−0.32 to −0.004] | −1.90 | 0.057 |

*Displacements received*: interaction received from another individual defined as above, causing recipient to raise head, move sideways or take a step away in any direction.

*Scratching*: Using either the mouth or the hoof to scratch the body.

*Startle response:* A quick reaction to an unexpected stimulus, the startle usually involved a quick movement, either jump backwards or sideways followed by looking up with ears pricked.

If any interruption lasted for over 1 min then the individual was classed as having stopped foraging. Note that individuals were only observed in detail when they were foraging, if an individual was not foraging when it was due to be observed, this was recorded (to calculate total foraging time, as described in "Animals and Ethical Statement") and but also counted as 'missed' in terms of recording interruptions. Once a missed individual was foraging again it was observed next as a priority (only if it had not yet already been observed for 20 min), but just for a single 5-min interval, before resuming the original order. This was to maximise the collection of data on foraging efficiency for each individual.

The frequency of foraging interruption (a proxy for foraging efficiency) was calculated as the number of instances of all interruptions per minute foraging. Separate frequencies were also determined for each interruption category (Table 1). The duration of interrupted foraging referred to the total percentage of time spent interrupted per individual.

## Dominance rank

Although the concept of dominance lacks universal explanatory power in describing social structure, it is a useful construct when considering the specific context of competition for a

limited food resource. Under such conditions, horses generally follow a linear ranking hierarchy, with occasional triangles and some influence of third-party interactions (*Houpt, Law & Martinisi, 1978*; *Van Dierendonck, De Vries & Schilder, 1995*; *Hartmann, Christensen & McGreevy, 2017*).

Here we defined dominance 'an asymmetry in the outcome of dyadic interactions between individuals, or a priority of access to resources' (*Drews, 1993*) and assessed it by measuring outcomes between dyadic pairs when feeding from hay feeders. Agonistic interactions were recorded continuously throughout the 3-h observation period (these were easily measurable alongside other observations). An agonistic interaction was defined as one individual approaching or displaying to another with the neck outstretched and ears back flat against the head and, crucially, the second individual moving away. Dominance rank was then calculated using the methods described by *Appleby (1980)*. The number of agonistic interactions both given and received was recorded for each herd individual, and then the number of other individuals that a focal individual both dominated and was dominated by was calculated.

Once an Appleby rank had been given, this was then adjusted to take into account herd size (as in *Giles et al. (2015)*). Adjusted dominance rank was calculated as $1 - (a - 1)/(h - 1)$, where $a$ is the Appleby rank and $h$ is the herd size. Where dominance rank or dominance status is referred to in this manuscript, this refers to this adjusted dominance rank.

## Body condition score

Measurements were taken immediately after the second set of observations on the herd had been completed. All study animals were accustomed to being handled. Body condition score was measured using the Henneke nine-point scale (*Henneke et al., 1983*) by a single trained observer (SLG). Six areas of the horse were scored between 1 and 9 and then averaged and rounded to the nearest 0.5, to obtain a single score. A score of five on the scale was taken to indicate an ideal body condition.

## Statistical analyses

Results were analysed using *Stata* 12.1 (Statacorp, College Station, TX, USA). Univariable relationships were assessed using mixed effects linear regression, the clustered study design was controlled for by including herd group and herd size as a random effects, on the basis that herd size or other herd specific factors such as environment could plausibly have some influence on foraging and interactive behaviours. Univariable relationships of primary interest were:

1. The relationship between dominance rank (adjusted for herd size) and body condition score.
2. The relationship between dominance rank (adjusted for herd size) and interruptions to foraging (as a proxy for foraging efficiency).
3. The relationship between body condition and interruptions to foraging (as a proxy for foraging efficiency).
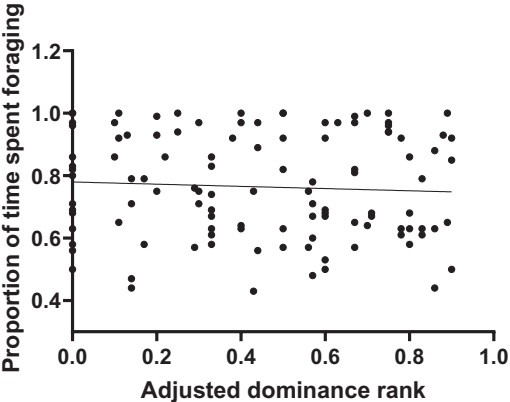

**Figure 1 Total foraging time as a function of adjusted dominance rank.** Foraging time was observed for the 116 study horses and expressed as a proportion of each individual's total time budget. Horses were kept in 20 discrete herds, and dominance rank was adjusted for herd group size. The plot shows a lack of association between dominance rank and total foraging time.

Following an initial univariable exploration of these relationships, relationships between the separate foraging interruption variables were also considered. In addition, breed, age, height, sex and whether or not the individual received supplementary feed were recorded as potential confounding variables. To be considered a potential confounder the variable had to be associated with both the explanatory and outcome variable, and not on the causal pathway between the two (*Petrie & Sabin, 2009*). Statistical significance was defined using $p \leq 0.05$ with a screening $p$-value for multivariable models of $p \leq 0.07$.

Mixed effects multivariable linear regression was then used to build a best-fit explanatory model for both adjusted dominance rank and body condition. The foraging interruption variables (see Table 1 for list) were added to the model one at a time, based on the strength of univariable association, starting with a minimal model. A likelihood ratio test was used to assess the contribution of each variable to the model fit and variables were retained on the basis of this and the adjusted $p$ value.

Multivariable analysis using a mixed effects linear regression model was also used to make predictions regarding interruptions to foraging—to explore whether this could be a possible mechanism linking dominance status and body condition. Duration of foraging interruption was associated with both dominance status and body condition, therefore this was added to a model containing adjusted dominance rank and body condition. Its explanatory contribution to the model was then assessed using both the adjusted $p$ and estimates and a likelihood ratio test.

## RESULTS

During 120 h of observation, the amount of time that individual animals spent foraging averaged 76.4% SD 0.17. Values per herd are given in Table S1. Figure 1 shows that there was no significant correlation between adjusted dominance rank and total foraging time ($r^2 = 0.004$, $n = 116$, $p = 0.51$) and Fig. 2 shows that there was no significant

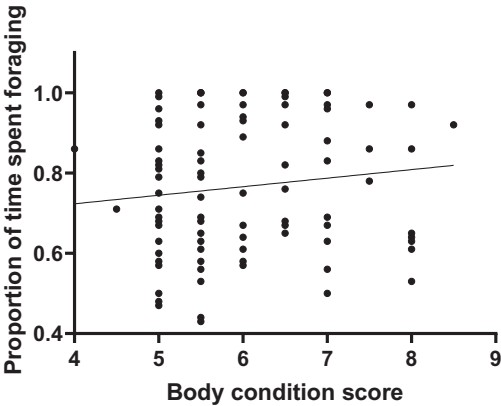

**Figure 2 Total foraging time as a function of body condition score.** Foraging time was observed for the 116 study horses and expressed as a proportion of each individual's total time budget. Body condition score was assessed using the Henneke 9-point scale. The plot shows a lack of association between body condition score and total foraging time.

correlation between body condition score (range 4–8.5) and total foraging time ($r^2 = 0.016$; $n = 116$, $p = 0.182$). This is important in thes interpretation of subsequent results.

## Univariable Analysis

### *The relationship between adjusted dominance rank and body condition score*

Adjusted dominance rank was positively associated with body condition score within our study population (Table 1).

### *Foraging efficiency*

During approximately 92 h of the 120 h total observation period, horses were foraging (total across all horses). During this time, the observed total numbers of each type of interruption contributing to foraging efficiency were: vigilance 2,518; movement whilst foraging 454; displacements given 198; displacements received 222; scratching 65; startle responses 5.

### *The relationship between dominance rank and foraging efficiency*

Although the frequency of foraging interruptions did not show evidence of association with adjusted dominance rank ($Z = -1.55$, $p = 0.12$, Table S2), the total duration of interruptions decreased as adjusted dominance rank increased (Table 1). An increase in adjusted dominance rank was also associated with a decrease in some specific interruption behaviours, namely instances of movement whilst foraging, displacements given, and displacements received (Table 1). Figure 1 shows that the reduced foraging efficiency of subordinate individuals is not compensated for by an increase in total foraging time.

### *The relationship between body condition score and foraging efficiency*

The number of incidences (frequency) of foraging interruptions occurring during foraging bouts was lower for animals with higher body condition scores. Vigilance decreased with an increase in body condition (Table 1), but none of the other separately defined

**Table 2 The final multivariable explanatory model for adjusted dominance rank, using mixed effects linear regression, controlling for herd group and herd size as random effects.**

| Explanatory variable | β | SE | 95% CI | Z | p |
|---|---|---|---|---|---|
| Frequency of being displaced | −2.71 | 0.35 | [−3.43 to −2.00] | −7.43 | <0.001 |
| Frequency of displacement towards others | 0.86 | 0.28 | [0.31–1.40] | 3.11 | 0.002 |
| Body condition score | 0.04 | 0.02 | [0.005–0.08] | 2.20 | 0.027 |
| Constant | 0.26 | 0.13 | [0.01–0.52] | 2.06 | 0.039 |

**Table 3 The final multivariable explanatory model for body condition score, using mixed effects linear regression, controlling for herd group and herd size as random effects.**

| Explanatory variable | β | SE | 95% CI | Z | p |
|---|---|---|---|---|---|
| Vigilance frequency | −0.89 | 0.30 | [−1.48 to −0.31] | −3.01 | 0.003 |
| Adjusted dominance rank | 0.63 | 0.29 | [0.06–1.18] | 2.19 | 0.029 |
| Constant | 6.14 | 0.23 | [5.68–6.59] | 26.55 | <0.001 |

foraging interruptions showed any association with body condition (Table S2). Figure 1 shows that the reduced foraging efficiency of individuals with lower body condition is not compensated for by an increase in total foraging time.

### Associations between the individual foraging interruption variables and consideration of potential confounders

Frequency of 'displacements received' was strongly associated with 'moving whilst foraging' and 'displacements given'. Frequency of 'displacements given' was also associated with 'moving whilst foraging' (Table 1).

In this study, none of the potential confounder variables (breed, age, height, sex) were associated with body condition score, adjusted dominance rank or any category of interrupted foraging, and there were no biologically plausible interactions, therefore adjusted estimates were not required. This also included whether or not a horse received additional supplementary feed, which showed no evidence of association with either adjusted dominance rank ($Z = −0.50$, $p = 0.61$) or body condition ($X_9^2 = 12.40$, $p = 0.19$).

## Multivariable analysis
### Model for adjusted dominance rank
Controlling for other model variables, frequency of 'displacements received', 'displacements given' and body condition score were associated with adjusted dominance rank (Table 2).

### Model for body condition score
Controlling for other model variables, vigilance frequency and adjusted dominance rank were strongly associated with body condition score (Table 3).

**Table 4 Multivariable linear regression model showing the effect of foraging efficiency (total duration of foraging interruptions) upon the relationship between dominance status and body condition.**

| Explanatory variable | β | SE | 95% CI | Z | p | Likelihood Ratio Test | |
|---|---|---|---|---|---|---|---|
| | | | | | | $\chi^2_1$ | p |
| Adjusted dominance rank | 0.55 | 0.29 | [−0.03 to 1.13] | 1.86 | 0.06 | 3.39 | 0.06 |
| Total duration of foraging interruptions | −0.07 | 0.04 | [−0.15 to −0.005] | −2.12 | 0.03 | 4.29 | 0.04 |
| Constant | 6.10 | 0.28 | [5.55–6.64] | 22.12 | <0.001 | – | – |

### *The relationship between body condition score and adjusted dominance rank when taking into account interruptions to foraging*

The association between body condition score and adjusted dominance rank was weaker when total duration of foraging interruptions (or time spent interrupted) was included in the model (Table 4, $p = 0.06$, as opposed to $p = 0.03$ in the univariable model). The effect size also reduced slightly (from a 0.66 increase in adjusted dominance rank per half unit of body condition score to 0.55). The likelihood ratio test results (Table 4) indicate that duration of foraging interruptions has a more significant contribution to the model fit ($p = 0.04$) than adjusted dominance rank ($p = 0.06$).

## DISCUSSION

The study explored the inter-relationships between foraging interruptions, dominance and body condition, controlling for herd size and herd identity effects. No effects of age, sex or height were detected in our study. Clearly, large horses have differing energy requirements from smaller ponies, whilst growing youngsters and older horses with reduced digestive efficiency (*Ralston, Squires & Nockels, 1989*) will also differ from young but mature adults. However, the horses in our study were housed in herds that contained animals of similar characteristics (see "Methods" and Table S3). For example, heavy horses were housed separately from lighter Thoroughbreds and smaller ponies. Although this policy greatly reduces or eliminates our ability to detect age and sex effects on foraging, it enhances our ability to detect the *relative* effects of dominance and body condition within herds. Importantly, our analysis showed that the relationships we detected applied across all herd types.

Within this study population, dominance status was positively associated with body condition, although this relationship was weaker when foraging efficiency was included in the multivariate model (Table 4). In addition, the association between body condition and foraging efficiency was stronger than that between body condition and dominance. Thus, whilst dominance explains some variation in body condition, our results highlight the potential role of factors other than social dominance that could influence foraging efficiency. Factors such as a tendency to show vigilance behaviour have been little explored to date but have the potential to greatly influence the ratio of energy gained vs energy expended during bouts of foraging.

There was no evidence that subordinate or low body condition individuals compensated for less efficient foraging by increasing total foraging time. Another recent study found

that horses with low body condition tend to adopt more passive behaviour (*Jorgensen et al., 2016*). Potentially such results may be due to a strong motivation to feed as a group in this species and thus synchronise feeding and resting behaviour (*Rands et al., 2008*). Subordinate or lower body score individuals were unlikely to remain foraging when conspecifics were not, supporting suggestions that social factors may result in stable differences in body condition within group living animals (*Rands, 2011b*; *Rands et al., 2008*). Indeed the tendency to synchronous feeding and resting (as in sheep, *McDougall & Ruckstuhl, 2018*) may be hard-wired as an adaptivebehaviour.

The lack of a compensatory change in total foraging time means that any variation observed in foraging efficiency could plausibly have an effect on body condition.

Given these results and previous theoretical predictions, an association between foraging efficiency, dominance and overall body condition was expected (*McNamara & Houston, 1990*; *Stillman et al., 2000*; *Rands et al., 2006*, *2008*) but our study is the first to explore the role of the different components of foraging efficiency, such as movement, social displacement or vigilance.

## Vigilance and body condition

Vigilance frequency was the individual interruption behaviour most strongly associated with body condition score—it showed a strong negative association. However, vigilance was not associated with dominance status. These results suggest that certain individuals may be more likely to conduct vigilance, perhaps on behalf of the group, regardless of their social status. These results do seem to support the suggestion that vigilance is an inherently costly activity (*Elgar, 1989*; *Fritz, Guillemain & Durant, 2002*; *Fattorini & Ferretti, 2019*; *Pacheco & Herrera, 1999*) as demonstrated by the negative association with body condition. However, lower body condition individuals may also be more stressed or nervous individuals, which would also explain the association with increased vigilance.

The complexity of vigilance as a single trait may somewhat explain the lack of observed association with dominance status. Vigilance may serve a range of functions in group living animals (*Fattorini & Ferretti, 2019*), including anti-predatory behaviour (*Elgar, 1989*; *Hunter & Skinner, 1998*), monitoring of other herd members and scanning the environment for resources (*Underwood, 1982*). Ungulate mammals that are unexposed to predation have been observed to greatly reduce their vigilance behaviour (*Hunter & Skinner, 1998*). Horses, unexposed to predation, may therefore show relatively low levels of vigilance, with reasons other than anti-predatory vigilance having a proportionally larger role.

Alongside the association between dominance status and body condition, the association between body condition and vigilance provides evidence of two separate behavioural traits associated with body condition in group living animals. Behavioural predictors of body condition have so far received little attention in horses (for exceptions, see *Ingólfsdóttir & Sigurjónsdóttir, 2008*; *Giles et al., 2015*) and may warrant continued investigation, especially as obese horses (BCS > 7) may show differences in activity and eating behaviour when compared to lean horses (BCS 4-5) (*Moore, Siciliano & Pratt-Phillips, 2019*).

### Dominance status, movement during foraging and displacement interactions

Subordinate horses showed more movement whilst foraging, and were (as expected) more likely to receive displacements. Indeed, statistical analysis revealed that displacement was strongly associated with movement during foraging in our study population, with subordinate animals forced to move foraging location. Theoretical models and empirical studies have proposed that subordinate individuals may be forced to foraging positions carrying a greater risk of predation (*Hamilton, 1971*; *Hemelrijk, 2000*). Future studies could examine whether subordinate animals showed increased vigilance specifically when in displaced locations, and during non-foraging periods.

Overall our results therefore appear to support predictions that displacement reduces foraging efficiency for the recipient (*Bautista, Alonso & Alonso, 1998*; *Stillman et al., 2002*). Valuable foraging time is wasted not only over the initial dispute, but also in relocating to a new foraging location. In contrast, dominant horses tended to interrupt their own foraging to displace others, but these interruptions tended to be of short duration, allowing the dominant animal to return quickly to foraging. As our study herds were feeding from hay feeders, potentially displacement and movement occurred more often than would occur during foraging on pasture, due to the artificially close proximity of herd members (*Hoffmann, Bockisch & Kreimeier, 2009*).

## CONCLUSIONS

These results are novel and exciting in that they present the first behavioural evidence confirming a broad body of influential theoretical work (*Marshal et al., 2012*; *Petit & Bon, 2010*; *Rands et al., 2003*; *2006*; *Rands, 2011b*; *Sueur et al., 2013*) linking condition and behaviour in a group-living species. Our results suggest (in line with model predictions) that differences in energetic reserves (body condition) can emerge simply via a reduction in energetic intake by subordinates when dominants are present. This hypothesis could be further tested in a future prospective study. One application of our work is that information on individual horse dominance status could be included as a relevant factor when addressing health problems associated with equine obesity (*Giles et al., 2014*; *Robin et al., 2015*; *Menzies-Gow, Harris & Elliott, 2017*).

## ACKNOWLEDGEMENTS

We thank the staff at Redwings Horse Sanctuary, especially Roxane Kirton and Anne Whitehorn for their kind help with conducting the dominance tests and for the use of their horses and ponies.

### Funding

Sarah Giles was supported by an Industrial CASE Studentship funded by Biotechnology and Biological Sciences Research Council [grant number BB/H01568X/1] and WALTHAM Petcare Science Institute. The WALTHAM Petcare Science Institute

contributed to the funding of the studentship for Sarah L. Giles and employs Patricia A. Harris.

## Grant Disclosures
The following grant information was disclosed by the authors:
Biotechnology and Biological Sciences Research Council: BB/H01568X/1.
WALTHAM Petcare Science Institute.

## Competing Interests
Christine Nicol is an Academic Editor for PeerJ. Pat Harris is an employee of WALTHAM Petcare Science Institute. The authors declare that they have no other competing interests.

## Author Contributions
- Sarah L. Giles conceived and designed the experiments, performed the experiments, analysed the data, prepared figures and/or tables, authored or reviewed drafts of the paper, and approved the final draft.
- Pat Harris conceived and designed the experiments, authored or reviewed drafts of the paper, and approved the final draft.
- Sean A. Rands conceived and designed the experiments, authored or reviewed drafts of the paper, and approved the final draft.
- Christine J. Nicol conceived and designed the experiments, prepared figures and/or tables, authored or reviewed drafts of the paper, and approved the final draft.

## Animal Ethics
The following information was supplied relating to ethical approvals (i.e., approving body and any reference numbers):

The University of Bristol Animal Welfare and Ethical Review Board approved this research as an observational study (University Investigation Number UB/10/049).

## Data Availability
The raw data from all 116 horses are available in Supplemental File.

## Supplemental Information
Supplemental information for this article can be found online at http://dx.doi.org/10.7717/peerj.10305#supplemental-information.

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
