# Peer review of "Foraging efficiency, social status and body condition in group-living horses and ponies"

_PeerJ, doi:10.7717/peerj.10305_

## Round 0.1 · original submission · Major Revisions

Firstly, my sincere apologies for the time taken on this manuscript; I had challenges getting reviewers for this manuscript, and the COVID situation complicated this further. I have now had a chance to review the manuscript and the reviews we received. You will see that the reviewers have recommended a major revision; while they see the value in your manuscript - there are several details that need attention. In particular, ensuring the aims, results and conclusions align well.

Reviewer 1 ·

Basic reporting

This is a very well researched study and written report.

Experimental design

The design was explained well and was based on previous work.

Validity of the findings

Good

Additional comments

Thank you for a very nice study in an interesting area!

Reviewer 2 ·

Basic reporting

The introduction is about a theoretical framework that is interesting. However the conclusions do not fit with the introduction or for that matter the findings.

I think the methods, data and analysis is presented in a clear way.

Experimental design

I am fine with the design and I think the number of horses is sufficient.

Validity of the findings

I think that the results are interesting but I would add more information about the vigilance behaviour.
If you want to keep the conclusions about welfare you need to explain the obesity issue already in the introduction and be more clear about differences between the obese horses and the others.

My main issue is around the conclusions.

Additional comments

In general I think the study is interesting but the manuscript needs more work.

Annotated reviews are not available for download in order to protect the identity of reviewers who chose to remain anonymous.

Reviewer 3 ·

Basic reporting

The subject of this manuscript is very interesting and fills a gap in the field. But I consider some methodological aspects of the study quite problematic.

Introduction
Line 37: definition of foraging would be appreciated to make the introduction clearer
Line 49: a more detailed description of the modelling framework proposed by Rands et al. in their different papers will be important to recall here. Indeed, the impact of social status on foraging that will be investigated in this manuscript is not stated in the current explanation.

Lines 87 to 93 – Aims: if a correlation is found between dominance rank and body condition (which is actually the case, see line 277) (i), the two following aims (ii and iii) need to be reconsidered. As a consequence, only one association with interruption of foraging (dominance rank or body condition*) has to be investigated contrary to the analyses that have been conducted (lines 287 to 296). Indeed, because these two characteristics are correlated, it is not surprising that the authors found that the total duration of interruptions decreased as adjusted dominance rank increased (lines 289-290) and that the number of incidences (frequency) of foraging interruptions occurring during foraging bouts was lower for animals with higher body condition scores (lines 296-298).

*because the two variables are correlated, I would prefer body condition since it is a more accurate individual measure than social rank. This will strengthen the results (cf. line 336-337).

Experimental design

- Lines 106-107: on my opinion, the sentence “The study explored the inter-relationships between foraging interruptions, dominance and body condition.” is unnecessary.
Moreover, the next sentences lines 107 to 110) will be better placed in the “Dominance rank” paragraph (line 203)
- Herd size between 2-10 individuals: the pressure from dominant individuals will be quite different in a group of 2 individuals (only one animal might displace the subordinate animal) than in a group with many individuals, where the pressure will be more important (a subordinate animal will be the target of more displacements/threats). This pressure might be estimated by a coefficient depending on herd size and then applied in the analysis.
- Line 124: the range of age raises some problems: young animals are not well situated in the hierarchy of a group. Moreover, these animals are in development and their body condition/physiology is certainly different from adults. The analysis will probably gain in robustness without these young individuals. Or their case might be considered separately. Moreover, older horses have a reduced digestive efficiency (See Ralston, S.L., Squires, E.L., Nockels, C.F. (1989). Digestion in the aged horse. J Equine Vet. Sci. 8(4), 203-205) that could influence the time spent foraging. It might be interesting to investigate this factor per se.
- What about the sex? If mares are pregnant or lactating, this will influence their body condition and energetic needs. More details are needed here.
- When I read the definition of Movement whilst foraging, I’m concerned by the fact that the two origins of such movement (displacement by another individual or simply changing of location at a walk) were not distinguished for subsequent analysis. Moreover, if a Movement whilst foraging is due to a displacement by another individual, it has to be recorded as a Displacements received. If this is the case, such event is accounted in two categories (and thus twice), which is incorrect. Logically, in the results section, the authors reported that they found a correlation between the frequency of ‘displacements received’ and ‘moving whilst foraging’ (line 304). I suggest reconsidering the classification of both behaviours.
- Dominance rank: I have no problem to infer Dominance rank from displacements emitted and received which is a classical method for measuring dominance in social groups but I don’t understand the justification (lines 212 to 217, see after in italics) that renders the understanding confusing: The number of displacements both given and received was recorded for each herd individual, and then the number of other individuals that a focal individual both dominated and was dominated by was calculated (see Appleby, 1980). Note that frequency of displacements is not the same as dominance rank. Herds may have very interactive individuals who are not particularly high or low ranking, displacing others and being displaced often, whereas the highest-ranking individual may actually displace others infrequently.

Validity of the findings

I’ve previously commented some results in the experimental design section.
Lines 317-318: If I well understand, the Model for adjusted dominance rank investigates whether the frequency of ‘displacements received’ and ‘displacements given’ are linked to the adjusted dominance rank. However, dominance rank was determined thanks to these two behaviours. This analysis sounds odd for me.

Additional comments

Discussion: It is difficult to comment this part because the concerns I’ve raised above will have consequences on the results and thus, on the interpretations. This is illustrated in the first paragraph of the Discussion.
Concerning the interpretation between vigilance and body condition, I think that recording vigilance when animals were not foraging to compare their rates of vigilance when they forage would have allowed to verify whether some animals were more naturally vigilant than others. Unfortunately, in the protocol, the observer stopped observing an animal when this latter stopped to forage for more than one minute. This point needs to be discussed.
Lines 375-377: the authors wrote: Theoretical models and empirical studies have proposed increased vigilance in subordinate individuals due to more risky foraging positions (Hamilton, 1971; Hemelrijk, 2000) but this was not supported here. I could be wrong, but I didn't find nor in the methods neither in the results section any record and analysis of these risky foraging position.
Conclusions: On my opinion, this section is more perspectives than a conclusion about the study.

---

## Round 0.2 · Minor Revisions

I have now had an opportunity to review your revised manuscript and the responses to review, and I am satisfied with the changes that have been made. One of the original reviewers has also reviewed the manuscript and has a few small suggestions for you. I agree that the conclusion should correspond clearly with the aims and what was discussed in the discussion without introducing new mater. In addition to the changes highlighted by the reviewer, I have a few suggested changes of my own:

line 239: two words appear to be unintentionally joined here.
Reference list: there are a few places where the author name has not been bolded - please correct this.

Tables - please ensure you use the same level of precision (number of decimal places) within each column (i.e. your p-values)

Figures - please present axis labels without underscores, and indicate units of measurement (e.g. Foraging_time should be Foraging time (mins)). Figure headings are not required, so please remove these.

Figure captions should be stand-alone (e.g. can be interpreted by someone who has not read your article) so please revise the figure captions so they are more informative.

Reviewer 2 ·

Basic reporting

Looks good to me

Experimental design

Good

Validity of the findings

Looks good

Additional comments

I think the MS has improved a lot and is very interesting. Journals vary in how the aims and conclusions correspond so I leave for the editor to decide, however I prefer conclusions that very clearly corresponds to the aims. In this case the aims are very clear and defined whereas the conclusions are more speculative and include a statement on the obesity issue in pets, horses and humans.

However this may be a matter of taste and style in the journal (adding a new aspect in the conclusions, without really discussing it before hand).

Apart from that I am happy with the MS now.

Annotated reviews are not available for download in order to protect the identity of reviewers who chose to remain anonymous.

---

## Round 0.3 · accepted · Accept

Thank you for making the changes to the article. I have reviewed the changes that were made, and the responses to review, and I have no further changes to request.